# Antimicrobial Wound Dressings: A Concise Review for Clinicians

**DOI:** 10.3390/antibiotics12091434

**Published:** 2023-09-11

**Authors:** Faraz Yousefian, Roksana Hesari, Taylor Jensen, Sabine Obagi, Ala Rgeai, Giovanni Damiani, Christopher G. Bunick, Ayman Grada

**Affiliations:** 1Goodman Dermatology, Roswell, GA 30076, USA; 2Lake Erie College of Osteopathic Medicine, Bradenton, FL 34211, USA; 3St. George’s University School of Medicine, West Indies P.O. Box 7, Grenada; 4College of Medicine, University of Arizona, Tucson, AZ 85724, USA; 5Hai Al-Andalus Primary Healthcare Center, Tripoli 13555, Libya; 6Italian Center of Precision Medicine and Chronic Inflammation, 20122 Milan, Italy; 7Department of Biomedical, Surgical and Dental Sciences, University of Milan, 20122 Milan, Italy; 8Department of Pharmaceutical and Pharmacological Sciences, University of Padua, 35131 Padua, Italy; 9Department of Dermatology, Yale University School of Medicine, New Haven, CT 06510, USA; 10Program in Translational Biomedicine, Yale University School of Medicine, New Haven, CT 06510, USA; 11Department of Dermatology, Case Western Reserve University School of Medicine, Cleveland, OH 44106, USA

**Keywords:** wound healing, infection prevention, wound management, chronic wounds, skin ulcers, dressing, antimicrobials, antibiotics, antiseptics, resistance

## Abstract

Wound management represents a substantial clinical challenge due to the growing incidence of chronic skin wounds resulting from venous insufficiency, diabetes, and obesity, along with acute injuries and surgical wounds. The risk of infection, a key impediment to healing and a driver of increased morbidity and mortality, is a primary concern in wound care. Recently, antimicrobial dressings have emerged as a promising approach for bioburden control and wound healing. The selection of a suitable antimicrobial dressing depends on various parameters, including cost, wound type, local microbial burden and the location and condition of the wound. This review covers the different types of antimicrobial dressings, their modes of action, advantages, and drawbacks, thereby providing clinicians with the knowledge to optimize wound management.

## 1. Introduction

Wound management is a significant challenge to healthcare systems worldwide, requiring a multidisciplinary approach. Chronic wounds, in particular, contribute a substantial economic burden, requiring prolonged and expensive interventions, multiple medical visits, and the potential for complications such as infections, amputations, or hospitalizations. The prevalence of chronic wounds is increasing due to the aging population and the rising rates of chronic diseases such as diabetes, cardiovascular disease, and obesity. This trend contributes to the economic burden of wound care on healthcare systems, third-party payors, and individuals.

A recent study underscored the considerable financial implications of wound care, documenting Medicare expenditures between USD 28.1 billion and USD 96.8 billion for various wound types. Notably, costs associated with outpatient services were higher, estimated between USD 9.9 billion and USD 35.8 billion, compared to inpatient costs, which ranged from USD 5.0 billion to USD 24.3 billion [1]. This information is particularly relevant in the context of dermatologic surgery, where many procedures are performed on an outpatient basis. Among the varied costs, surgical wounds represented the most substantial expenses, particularly when considering the management of infected wounds. Infections are a common and concerning complication of cutaneous wounds, potentially leading to delayed healing, and in severe cases, sepsis and death. When infections coincide with the ongoing immune response of the initial wound, inflammation is prolonged due to excessive tissue damage, culminating in delayed and impaired wound repair [2]. Studies have demonstrated a significant correlation between a wound’s microbial bioburden and its healing trajectory [3]. Further, the persistence of wound infections significantly contributes to wound chronicity [4], underscoring the potential of antimicrobial dressings as a strategy to mitigate wound infections and enhance healing outcomes [2,5]. Bacterial presence in a wound can be categorized into three distinct situations: (a) contamination: the presence of bacteria on the surface without multiplication, and the absence of clinical disease; (b) colonization: the bacteria multiply, but without signs and symptoms of infection; and (c) infection: the proliferation of bacteria associated with local host reaction, delayed healing, and tissue damage [6]. Infected wounds clinically present with erythema, warmth, edema, and pain or local tenderness. Increased wound drainage, purulence, and new or worsening malodor can occur. Antimicrobial dressings are indicated when critical colonization or localized infection is suspected. Systemic signs such as fever and leukocytosis are indicators of progression to bacteremia or septicemia. In such cases, systemic antibiotics are warranted. Wound dressings serve a dual purpose: fostering an optimal moisture environment conducive to re-epithelialization and acting as a physical barrier against microbial penetration, colonization, and proliferation within the wound and the dressing itself [2,5]. Various dressing forms, including gauze, film, hydrocolloid, hydrogel, foam, alginate, and antimicrobial dressings, have been utilized in wound care, each exhibiting unique properties and clinical benefits. Antimicrobial wound dressings employ disinfectants, antiseptics, or antibiotics to reduce and eliminate local wound bioburden [5]. This review covers the different types of antimicrobial dressings, active ingredients, mechanisms of action, advantages, and drawbacks, thereby providing clinicians and researchers with the basic knowledge to optimize wound management. A literature review of electronic databases, including PubMed and Google Scholar, was conducted to identify articles written in the English language and published between 1970 and 2023. The search utilized keywords such as “wound healing”, “infection prevention”, “wound management”, “antimicrobial dressings”, and other relevant terms. Relevant publications were carefully examined for supplementary information. Peer-reviewed articles, systematic reviews, and meta-analyses were included to ensure the inclusion of current and high-quality evidence.

## 2. Types of Antimicrobial Dressings

### 2.1. Direct Topical Application of Antiseptic

Topical antiseptics can be categorized into different classes depending on their mechanism of action. These include emulsifiers, oxidizers, acids, heavy metals, alcohols, aldehydes, anilides, bisphenols, and phenols [7]. The direct application of these topical antiseptics on wound beds, coupled with appropriate dressings, is generally limited to short durations to mitigate the risk of tissue damage [5,8]. Frequently used topical antiseptics include hydrogen peroxide, Dakin solution (a dilution of bleach in water), Eusol solution (a mixture of diluted bleach in water with boric acid), and diluted acetic acid (a combination of vinegar and water). These antiseptics are typically applied topically for wound cleansing purposes [2,5]. Vashe solution is a saline-based wound cleanser that contains hypochlorous acid (HOCl) whereas Dakin solution must first undergo a reaction with water to form HOCl. However, both of these solutions share a similar mechanism of action, functioning as bactericidal and fungicidal agents. A recent retrospective analysis found that the addition of HOCl into dressings used in the care of venous leg ulcers led to the elimination of biofilms and promoted the complete closure of the ulcers [9].

### 2.2. Silver-Based Wound Dressings

Silver is recognized in dermatology for its clinical effectiveness, particularly when applied as silver sulfasalazine in the treatment of burn-related wounds. The introduction of silver-impregnated dressings in forms such as foams, hydrofibers, and hydrocolloids has expanded its utility in the field to encompass a broader range of wound types, whether they are colonized or infected (Table 1) [6]. Notably, silver-based preparations are being utilized in burns, ulcers, surgical wounds, and chronic wounds [9]. Silver is an example of a heavy metal antiseptic. It possesses broad-spectrum bactericidal activity covering both Gram-positive and Gram-negative bacteria that can be used in heavily-infected wounds and when the presence of drug-resistant bacteria is suspected [5]. Specifically, Silver exhibits cytotoxic effects by inducing damage to various cellular organelles essential for bacterial gene transcription and cell wall synthesis. This includes DNA/RNA, mitochondria, and enzymes [2,5,10]. The slow release of silver ions reduces bacterial contamination while minimizing potential cytotoxic effects on healthy tissues. A variety of silver dressings (e.g., Acticoat™, Actisorb^®^ Silver, Contreet Foam, Contreet Hydrocolloid, and Silverlon™) may provide 3–7 days of antimicrobial effect and reduce hospitalization time, especially in scenarios of wound-associated sepsis and bacteremia [11]. Silver dressings may inhibit the host fibroblast activity required for wound healing due to its cytotoxic property; therefore, it is not recommended for infection prophylaxis. Silver dressings are safe to remain on patients requiring magnetic resonance imaging (MRIs) for soft-tissue and bone structures [12].

### 2.3. Iodine-Based Wound Dressings

Iodine has been recognized for its capacity to reduce the microbial burden in chronic wounds.^13^ In clinical practice, it is predominantly utilized in two distinct forms: povidone-iodine and cadexomer iodine [13]. Povidone-iodine has proven effective against bacteria, viruses, fungi, spores, protozoa, and amoebic cysts [14,15,16,17,18]. It exhibits potent antimicrobial activity due to its strong oxidative effects on the functional groups of amino acids and fatty acids. It predominantly interacts with the amino (-NH2) and thiol (-SH) groups in amino acids, and the carbon–carbon double bonds in fatty acids. This interaction with iodine leads to rapid structural and functional damage to bacterial and fungal cells, thereby inhibiting their growth and survival [19]. In addition to antimicrobial effects, povidone-iodine has also demonstrated anti-inflammatory properties as it scavenges radical oxygen species [19,20,21,22,23]. Cadexomer iodine dressing functions through a dual mechanism: absorption of exudates and controlled release of iodine. When applied to a wound, the highly absorbent cadexomer matrix within the dressing soaks up wound exudates, expands to form a gel, and aids in the debridement of the wound. Concurrently, as the dressing absorbs fluid, it releases iodine slowly. Iodine, a potent antimicrobial, combats a broad spectrum of pathogens including bacteria, fungi, viruses, and yeasts, thereby ensuring a sustained antimicrobial effect while minimizing potential iodine toxicity. The gel formed through the absorption of exudates facilitates the cleaning of the wound bed by physically removing debris and bacteria upon dressing removal. In addition, iodine’s anti-inflammatory properties help reduce inflammation and promote healing. Lastly, by absorbing exudates and debris, the dressing decreases wound size and depth, further aiding in the healing process [9]. A recent meta-analysis suggests that cadexomer iodine dressings are associated with better healing outcomes when compared to the standard of care for venous leg ulcers [24].

Compared to silver, iodine demonstrates less cytotoxic activity and offers a more cost-effective option, making it a useful prophylactic choice for wounds with a high risk of infection [5]. Different forms of iodine, including cadexomer iodine and povidone-iodine, can be incorporated into occlusive dressings such as hydrocolloids (e.g., Iodosorb^®^) and hydrogels (e.g., Iodoflex^®^) [11]. A recent study comparing the healing rates of leg ulcers treated with various dressings—hydrocolloid, povidone-iodine, silver sulfadiazine, and chlorhexidine digluconate—found that ulcers treated with povidone-iodine exhibited significantly improved healing rates and shorter healing times [9]. Despite these benefits, the use of iodine-based dressings does carry a risk of systemic iodine absorption, which can potentially lead to thyroid dysfunction. Consequently, these dressings are not recommended for children, pregnant and lactating women, or patients with a history of thyroid dysfunction or iodine sensitivity [5].

The Wolff–Chaikoff effect, a decrease in thyroid hormone production due to excessive iodide, must be considered when using iodine-based wound dressings. Although multiple studies indicate no significant adverse effects or meaningful changes in thyroid hormone levels from various iodine dressings, isolated instances of transient hypothyroidism have been reported after the use of iodoform gauze [25,26,27]. Therefore, monitoring thyroid function is recommended for all patients treated with iodine-based dressings, particularly those with a higher susceptibility to thyroid dysfunction.

### 2.4. Biguanide-Based Wound Dressings

Polyhexamethylene biguanide (PHMB) is an antiseptic often incorporated into wound dressings. As a member of the biguanide family, it consists of a mixture of polymers and acts primarily as a disruptor of microbial cell walls and membranes. While it is known to interfere with intracellular targets like chromosomes, it is not the only antiseptic with such capabilities [7,9]. PHMB is effective against a wide variety of bacteria, including both Gram-positive and Gram-negative strains, with notable efficacy against methicillin-resistant Staphylococcus aureus (MRSA) and vancomycin-resistant Enterococci (VRE). Moreover, PHMB exhibits antimicrobial activity against fungi such as Candida and Aspergillus species, amoeboids like the Acanthamoeba species, and both enveloped and non-enveloped viruses [28,29,30]. Despite its broad-spectrum effectiveness and widespread use, no instances of microbial resistance to PHMB have been reported. This antiseptic is incorporated into several commercially available wound dressings, including ActivHeal^®^ PHMB, Excilon™ AMD, Telfa™ AMD, Kerlix™ AMD, and Kendall™ AMD. A study has shown that using PHMB foam dressing can significantly reduce bacterial burden, polymicrobial organisms, wound pain, and wound size in ulcer treatment compared to similar non-antimicrobial foam dressing [9].

### 2.5. Antibiotics in Wound Dressings

Various classes of antibiotics, including aminoglycosides, beta-lactams, glycopeptides, quinolones, sulfonamides, and tetracyclines, have been incorporated into wound dressings [31]. Each antibiotic class targets bacteria in unique ways: beta-lactams and glycopeptides inhibit cell wall synthesis; aminoglycosides and tetracyclines interfere with protein synthesis; sulfonamides inhibit nucleic acid synthesis; and quinolones inhibit DNA replication and transcription [31]. For instance, the antibiotic mupirocin is effective against Gram-positive bacteria like MRSA, and metronidazole works well against anaerobic bacteria [2]. However, the long-term or incorrect use of broad-spectrum topical antibiotic wound dressings can foster multidrug-resistant bacteria [32,33]. It is reported that over 70% of bacteria causing wound infections are resistant to at least one common antibiotic [31]. Notably, there is a growing prevalence of mupirocin-resistant Staphylococcus aureus strains, reducing mupirocin’s effectiveness in preventing invasive infections, despite it being the only approved antibiotic for MRSA decolonization [34]. Therefore, considering culture and sensitivity tests is vital to choosing the right antibiotic for an infected wound. Surface swab cultures often have limited utility due to the presence of transient bacteria on wounds and skin. However, quantitative swab cultures can aid in identifying the pathogenic organism, with an infection confirmed at 1 × 10^6^ organisms per gram of tissue [6].

### 2.6. Other Antiseptic Agents Utilized in Wound Dressings

#### 2.6.1. Medical-Grade (Manuka) Honey

Long revered as a natural healer, honey’s medicinal value has been reaffirmed through modern scientific research and its incorporation into antimicrobial wound dressings [35,36], including MediHoney^®^, Activon Tulle^®^, Algivon^®^, and Actilite^®^ [36]. Honey-based dressings inhibit bacterial growth due to their osmotic effect, acidic pH, and the presence of antibacterial substances like methylglyoxal [37,38,39,40]. Manuka honey (MH), in particular, retains antibacterial activity even in biological fluids due to its non-peroxide component [35,36]. Impressively, honey demonstrates inhibitory effects on over 50 strains of bacteria without indications of microbial resistance [41,42].

In addition to its antimicrobial benefits, honey provides topical nutrition to wounds and promotes healing. In patients with venous ulcers, MH dressings have been associated with significant reductions in wound size, pain, and malodor [43]. MediHoney^®^, through its osmotic effects, facilitates debridement by drawing fluid from deep tissue layers, promoting the removal of devitalized tissues. Honey’s anti-inflammatory properties and stimulation of angiogenesis, granulation, wound contraction, and epithelialization aid the wound healing process [35,36]. 

In a study comparing the healing process of diabetic foot ulcers treated with MediHoney Tulle Dressing versus conventional saline-soaked gauze, MediHoney dressings led to rapid bacterial clearance, decreased antibiotic need, and reduced hospitalization time. Patients treated with MediHoney had an average healing time of 31 +/− 4 days, compared to 43 +/− 3 days with conventional dressings, and over 78% demonstrated sterile wounds within a week, compared to 35.5% with conventional dressings [44]. These findings further substantiate the multifaceted therapeutic potential of honey in wound management.

#### 2.6.2. Plant-Derived Natural Compounds

Essential oils, plant-derived compounds with diverse properties, have been integrated into antimicrobial wound dressings. These oils offer several benefits in wound care due to their antibacterial, antiviral, antifungal, analgesic, anti-inflammatory, and antioxidant effects [45,46,47]. Each type of oil can also have unique properties that enhance its effectiveness in wound care [48,49]. For example, oregano, known for its antimutagenic effects, has been incorporated into cellulose acetate fibers to improve the efficacy of antimicrobial dressings. Similarly, tea tree oil is used in Burnaid^®^ hydrogel dressings to treat burns in several countries outside the U.S. St John’s Wort promotes skin re-epithelization, while lavender accelerates the formation of granulation tissue when applied topically [50,51,52]. Notably, essential oils have a low tendency to promote microbial resistance compared to traditional antibiotics [53]. This characteristic makes them a valuable tool against multidrug-resistant bacteria in wound infections, offering a potential alternative or supplement to conventional antibiotic therapy [54,55]. Lastly, polyphenols are components of plants that have been shown to have antimicrobial activity against broad-spectrum bacteria and fungi [56,57]. Polyphenol has one or several phenolic groups and several studies have shown these compounds, especially flavonoids, exert their antimicrobial activity through augmenting antibiotic activity and direct microbial elimination and attenuation [57].

#### 2.6.3. Nanoparticles

Nanoparticles (NPs) have garnered significant attention in regenerative medicine as a promising alternative to traditional antibiotic therapy for treating multidrug-resistant bacterial wound infections. They possess advantageous physicochemical, biological, and optical properties, making them well-suited for various biomedical applications, particularly antimicrobial wound dressings [56,57]. NPs can be categorized into two main types: metallic and non-metallic. Non-metallic NPs can further be divided into organic NPs and carbon-based NPs. Metallic NPs include gold (Au), silver (Ag), platinum (Pt), copper oxide (CuO), iron oxide (Fe_3_O_4_), and zinc oxide (ZnO) [58]. The antimicrobial activities of metallic NPs are primarily attributed to their large surface areas, unique particle shapes, and small sizes. Additionally, their ability to generate reactive oxygen species contributes to their antimicrobial activity [59]. 

On the other hand, non-metallic NPs consist of organic NPs such as dendrimers, ferritins, micelles, liposomes, and polymer NPs, while carbon-based NPs include fullerenes, graphene, carbon black, carbon nanofibers, and carbon nanotubes (CNTs), and occasionally activated carbon [60]. The antimicrobial activity of carbon-based NPs is closely linked to their size and surface area, with smaller sizes and larger surface areas showing higher antimicrobial activity [61,62,63].

However, it is essential to consider potential side effects and risks associated with excessive NP exposure. One concern is the dispersion and accumulation of NPs in different organs of the body, including the brain, lungs, kidneys, and skin, which may trigger toxic reactions within the host [64]. To address these risks, conducting in vivo studies on the biodistribution and safe degradation profile of NPs before their clinical application in antimicrobial wound dressings is crucial. By thoroughly investigating the in vivo behavior of NPs, we can better understand the potential risks and develop safer and more effective antimicrobial wound dressings [62]. 

#### 2.6.4. Chitosan-Based Dressing

Chitin is a polysaccharide commonly found in the exoskeletons of arthropods and insects. The deacetylation of chitin results in the formation of chitosan, which exhibits broad-spectrum antimicrobial activity against both bacteria and fungi. Chitosan’s antimicrobial mechanisms of action can be classified as extracellular, intracellular, or both, depending on the type of microorganism and the specific chemical properties of the chitosan. For example, high-molecular-weight chitosan is impermeable to the cell membrane or cell wall. As a result, it is hypothesized to act as a metal chelator and to disrupt nutrient passage. Additionally, it may alter the physicochemical characteristics of the cell membrane in the extracellular space. Conversely, low-molecular-weight chitosan can influence intracellular processes, affecting DNA, RNA, and mitochondrial functions [65]. Chitosan can be incorporated into a range of wound dressings to leverage its antimicrobial and biocompatible properties. These dressings include hydrogels, which provide a moist wound environment; films that offer a breathable protective layer; absorbent sponges tailored for wound cavities; nanofibers, mirroring the natural extracellular matrix; foams designed for high exudate absorption; particles and beads to deliver therapeutic agents; membranes forming a protective shield; and composite dressings that combine chitosan with other beneficial materials for optimal wound care [65]. However, the antimicrobial efficacy of chitosan and its derivatives has been found to be considerably lower than that of conventional antimicrobial products. Therefore, further studies are essential to evaluate the potential of chitosan in antimicrobial wound dressings.

#### 2.6.5. Antimicrobial-Peptide-Based Dressings 

Antimicrobial peptides (AMPs), also referred to as host defense peptides (HDPs), are bioactive molecules found in various living organisms, playing crucial roles in their defense mechanisms. These peptides serve as the first line of defense in the innate immune system against bacteria, fungi, and viruses. AMPs carry out their microbicidal activity through both membrane-targeting and non-membrane-targeting mechanisms. In membrane-targeting mechanisms, AMPs integrate and accumulate within the membrane structure, subsequently compromising its functional integrity. Conversely, in non-membrane-targeting mechanisms, AMPs penetrate the cell either directly or through endocytosis, inhibiting processes like protein synthesis, nucleic acid biosynthesis, protease activity, and cell division. Besides their broad-spectrum microbicidal properties, AMPs also promote wound healing by stimulating angiogenesis, cytokine release, cell migration, and the proliferation of dermal cells [66,67,68]. At present, numerous preclinical and clinical trial studies focus on AMPs for infectious diseases. The FDA has approved several products containing or inspired by AMPs for skin or wound bacterial infections, including Neosporin^®^ (which contains gramicidin), Dalvance™ (dalbavancin), Cubicin^®^ (daptomycin), Orbactiv^®^ (oritavancin), and Vancocin^®^ HCl (vancomycin) [66,67,68].

**Table 1 antibiotics-12-01434-t001:** List of different types of antimicrobial wound dressings based on active ingredient.

Active Ingredient	Antimicrobial Properties	Dressing Forms	Uses	Precautions	Examples of Dressings
Silver	- Broad-spectrum antimicrobial - Bactericidal [6,69]	- Alginates - Foams - Hydrophilic fibers - Gels - Powders - Impregnated gauze - Combined with oxidized regenerated cellulose/collagen - Combined with collagen - Coated polyethylene mesh - Impregnated hydrocolloids - Combined with charcoal in a sachet	Superficially infected wounds, burns, and ulcers [6]	Cytotoxicity, older formulations rapidly inactivated necessitating frequent reapplication	Acticoat™ Actisorb^®^ SilverContreet Foam Contreet Hydrocolloid Silverlon™
Nanoparticles: Metals and metal oxides (silver, zinc oxide, iron oxide, cerium dioxide, titanium dioxide). Non-metals (dendrimers, ferritins, micelles, liposomes)	- Broad-spectrum antimicrobial activity - Bactericidal [70]	- Hydrogel - Hydrocolloid	Burns, pressure ulcers	Dispersion and accumulation in different organs of the body, leading to toxicity [62,64,71]	Acticoat^®^ Aquacel Ag^®^ Silvasorb^®^
Iodine	- Broad-spectrum antimicrobial - Bactericidal and fungicidal [19]	- Iodophor-impregnated gauze - Slow-release molecular iodine in cadexomer starch beads - Povidone-iodine-impregnated non-adherent dressing	Superficially infected wounds [6] Wounds with risk of infection [6]	Local tissue toxicity and irritation [6] Long-term exposure may impact thyroid function [5,72]	Iodosorb^®^, Iodoflex^®^
Gentian violet and methylene blue (GV/MB)	- Broad-spectrum antimicrobial activity [7]. GV and MB organic dyes have oxidation–reduction (redox) potentials in the range of many electron transport components of oxidative metabolism	- Polyurethane foam	Colonized and critically colonized wounds with varying levels of exudate	Contraindicated for third-degree burns	Hydrofera Blue^®^
Biguanides: Polyhexamethylene biguanide (PHMB), chlorhexidine	- Cationic emulsifier and broad-spectrum antimicrobial [7] - Bactericidal, virucidal, cysticidal [28,29] - Promotes tissue granulation and wound healing [7]	- Ribbon gauze - Gauze squares - Transfer foam - Backed foam - Non-adherent - Gels	Burns [7] Critically colonized and infected chronic wounds [64]	Possibly cytotoxic. Repeated prolonged exposure at >2% may cause sensitization [65,73]	ActivHeal^®^ PHMB ExcilonTM AMD Telfa™ AMD Kerlix™ AMD Kendall™ AMD
Honey	- Bactericidal [66] - Peroxide and non-peroxide antibacterial activity [35,36,74] - Osmotic effects, acidic pH, presence of inhibitory substances (e.g., methylglyoxal) [37,38,39,40]	- Liquid form - Alginate pads - Hydrocolloids	Superficial and partial thickness burns [67]	Non-medical-grade honey products should be avoided, as they may harbor viable clostridium spores and exhibit uncertain antibacterial properties [67]	MediHoney^®^ Activon Tulle^®^ Algivon^®^ Actilite^®^
Plant-derived natural compounds (Oregano, Tea Tree Oil, St. John’s Wort, Lavender)	- Broad-spectrum antimicrobial - Bactericidal, insecticidal, analgesic, anticancer, antioxidant, and anti-inflammatory effects [51,75]	- Hydrogel	Burns [51] Surface infections [68]	Frequent application and/or the use of high concentrations may be necessary [76]	Burnaid^®^
Chitosan	- Broad-spectrum antimicrobial - Extracellular: cell membrane destabilizer - Intracellular: inhibitor of DNA, RNA, and mitochondrial processes	- Hydrogel - Hydrocolloid - Sponge - Film [77]	First- and second-degree burns Chronic wounds with high risk of infection	Allergic reaction to chitin in individuals with shellfish allergy	Tegasorb^®^ Chitoflex^®^ Chitoseal^®^ HemCon^®^
Antimicrobial peptides (AMPs)	- Broad-spectrum antimicrobial [78]	- Hydrogel - Hydrocolloid - Sponge - Film - Foam - Alginate - Silicone - Collagen	Infected wounds Biofilms Surgical wounds Chronic wounds Burns	Some AMPs might be sensitive to light, heat, or moisture. Proper storage conditions are crucial to maintain their efficacy. Cytotoxicity at higher concentration	Neosporin^®^ (gramicidin) Dalvance™ (dalbavancin) Cubicin^®^ (daptomycin) Orbactiv^®^ (oritavancin) Vancocin^®^ HCl (vancomycin)

## 3. Practical Considerations When Selecting an Antimicrobial Wound Dressing

Clinicians must carefully evaluate various factors pertaining to the wound when selecting an appropriate antimicrobial dressing. Key considerations include the wound’s size, depth, presence of necrotic tissue, foreign material, and the level and type of exudate. Assessing for existing infection or risk factors for infection is crucial. Based on the wound’s specific properties, clinicians may choose a dressing that maintains a moist environment, promotes absorption, or offers breathability. Practical aspects such as ease of application, maintenance, and removal, as well as the frequency of dressing changes and associated costs, should be taken into account. Additionally, patient preferences play a significant role. Factors like patient comfort, compliance, medical history, and allergies should be considered. Furthermore, the wound healing process can be influenced by the patient’s cardiovascular, nutritional, and immunological status, as well as their psychosocial and occupational factors. Certain medical conditions, including chronic inflammatory disorders, diabetes, vascular insufficiency, nutritional deficiencies, neurological defects, advanced age, and local factors such as pressure, infection, and edema, may hinder proper wound healing. Therefore, addressing these chronic comorbidities alongside wound care often requires a multidisciplinary approach.

Monitoring the wound healing process is vital to identify signs of complications, particularly infections. In cases where systemic symptoms of infection are present, systemic antimicrobial agents and debridement may be necessary. By taking all these factors into account and tailoring the wound dressing approach accordingly, clinicians can optimize wound healing and minimize the risk of complications.

## 4. Future Directions

Additional studies should be conducted to bridge the evidence gap on efficacy, tolerability, and safety profiles of individual antimicrobial compounds. Emerging areas of development for antimicrobial wound dressings include the integration of nanotechnology for targeted drug delivery, the development of biodegradable and bioactive materials to stimulate tissue regeneration, and the exploration of combination therapies with growth factors and antimicrobial peptides. Smart dressings with sensors may enable real-time wound monitoring, while 3D printing technology allows for personalized dressing designs. Advancements in immunomodulatory dressings, advanced delivery systems, and biofilm management aim to optimize wound healing and infection prevention. Additionally, research may lead to personalized wound care based on genomics and precision medicine, while a focus on environmental sustainability drives the use of eco-friendly materials in wound dressing development. Ongoing research and clinical validation will be crucial for translating these innovations into effective wound care practices.

## Data Availability

Not applicable.

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
