# Peer review of "Antimicrobial Wound Dressings: A Concise Review for Clinicians"

_antibiotics, 2023, doi:10.3390/antibiotics12091434_

Round 1

Reviewer 1 Report

In this manuscript, authors summarized different categories of antimicrobial wound dressings and provided commercial examples for each category. This article will be a good contribution to the field, especially to medical practitioners with all the examples provided. I suggest minor revision and please see below for further comments.

Comments:

1. The novelty of this work is not significant. The title is too broad, and as a result, I could easily find many review articles on this topic. I suggest authors give credits to these previous review articles by citing and mentioning them in the body text. And a clear statement on what novelty this manuscript can provide comparing to existing literatures is suggested. A title that clearly defines the scope of this review paper can be helpful, as well.

2. The section structures are not clear enough. Authors stopped numbering sections or subsections after Section 2.

3. The section “2. Method” is not very common for review papers since most review papers should be written in this way. If authors don’t have specific reasons to emphasize the methodology, I suggest removing this section.

4. Reference 16 looks extremely suspicious. I cannot find any legit online record for this article, except for a self-uploaded pdf on ResearchGate. I personally do not encourage people to produce or reuse trash in academia. I will leave authors to decide if this article is legit or valuable enough to be cited here.

Reviewer 2 Report

Here, Yousefian et al. present a comprehensive review of antimicrobial dressings, highlighting their potential in addressing the challenges of wound management, such as infection and delayed healing, with the aim to provide insights to guide clinicians in optimizing wound care strategies.

However, the quality of the review is very poor and of no use to readers. Few of the major issues are listed below:

1) With this broad topic, the authors can write at least 100 pages of review article by discussing at least 350-400 research papers. The first mistake that the authors made is writing a review article for a broad topic that could have been presented as a textbook. I suggest authors to chose a specific topic and then write a review on it.

2) While the present topic could fill a textbook, the information provided in this review article spans less than 8 pages. Consequently, it resembles a blog post rather than a comprehensive review article.

Therefore, this article cannot be improved through major revision; instead, it needs to be completely rewritten, focusing on a specific topic that spans at least 20 pages to provide substantial value to the readers.

Reviewer 3 Report

The review under consideration admirably delves into various aspects of antimicrobial dressings, including their classification, mechanisms of action, benefits, and limitations, thereby equipping clinicians with vital insights to enhance wound management. However, it is evident that the discussion pertaining to antimicrobial materials remains somewhat oversimplified, leaving room for a more comprehensive exploration.

The review notably omits the inclusion of diverse antimicrobial materials, such as plant-based antimicrobial agents, polyphenols, chitosan, peptides, enzymes, and stimuli-responsive-based antimicrobial materials. These materials have garnered increasing attention in recent research and clinical applications, and their absence from the current review limits its potential to provide a truly encompassing understanding of contemporary wound management practices.

Furthermore, the absence of an illustrative figure showcasing an exemplar antimicrobial dressing is a notable gap in the presentation. Visual aids can significantly augment the comprehension of complex concepts and facilitate the translation of theoretical knowledge into practical applications.

To fortify the scholarly merit of this review, I recommend augmenting the discussion on antimicrobial materials to encompass the aforementioned types, thus fostering a more comprehensive and updated overview of available options. Additionally, the incorporation of a well-crafted figure depicting an example of an antimicrobial dressing would undoubtedly enrich the reader's engagement and enhance the overall educational value of the review. By addressing these points, the review can more effectively serve its intended audience of clinicians seeking informed guidance in the realm of wound management.

Round 2

Reviewer 2 Report

The authors have made a substantial efforts to address my queries.

Reviewer 3 Report

It's quite surprising to learn that the authors claim they don't have enough time to deliver a thorough review complete with systematic illustrations and exemplary figures. I find it challenging to accept such an irresponsible approach. While I can comprehend such constraints when discussing a research paper, what limitations are there when it comes to a review? Is the primary goal here to expedite the publication process?